# Biofilm in Endodontics: In Vitro Cultivation Possibilities, Sonic-, Ultrasonic- and Laser-Assisted Removal Techniques and Evaluation of the Cleaning Efficacy

**DOI:** 10.3390/polym14071334

**Published:** 2022-03-25

**Authors:** Uros Josic, Claudia Mazzitelli, Tatjana Maravic, Ales Fidler, Lorenzo Breschi, Annalisa Mazzoni

**Affiliations:** 1Department for Biomedical and Neuromotor Sciences, University of Bologna-Alma Mater Studiorum, 40139 Bologna, Italy; uros.josic2@unibo.it (U.J.); claudia.mazzitelli@unibo.it (C.M.); tatjana.maravic@unibo.it (T.M.); ales.fidler@mf.uni-lj.si (A.M.); 2Dental Clinic, Faculty of Medicine, University of Ljubljana, 1000 Ljubljana, Slovenia; annalisa.mazzoni@unibo.it

**Keywords:** biofilm, endodontics, irrigation, sonic, ultrasonic, laser

## Abstract

Incomplete and inadequate removal of endodontic biofilm during root canal treatment often leads to the clinical failure. Over the past decade, biofilm eradication techniques, such as sonication of irrigant solutions, ultrasonic and laser devices have been investigated in laboratory settings. This review aimed to give an overview of endodontic biofilm cultivation methods described in papers which investigated sonic-, ultrasonic- and Er:Yag laser-assisted biofilm removal techniques. Furthermore, the effectiveness of these removal techniques was discussed, as well as methods used for the evaluation of the cleaning efficacy. In general, laser assisted agitation, as well as ultrasonic and sonic activation of the irrigants provide a more efficient biofilm removal compared to conventional irrigation conducted by syringe/needle. The choice of irrigant is an important factor for reducing the bacterial contamination inside the root canal, with water and saline being the least effective. Due to heterogeneity in methods among the reviewed studies, it is difficult to compare sonic-, ultrasonic- and Er:Yag laser-assisted techniques among each other and give recommendations for the most efficient method in biofilm removal. Future studies should standardize the methodology regarding biofilm cultivation and cleaning methods, root canals with complex morphology should be introduced in research, with the aim of simulating the clinical scenario more closely.

## 1. Introduction

The elimination of pathogen microorganisms is routinely performed in conventional daily dental practice, being imperative for a predictable endodontic therapy [1,2,3,4,5]. The persistence of intraradicular infection due to incomplete bacteria removal or pathogen’s leakage within the canals is the most common cause of failure of root canal therapy. Enterococcus faecalis (*E. faecalis*), a Gram-positive, facultative anaerobic microorganism is most frequently detected in unsuccessful root canal therapy, with a prevalence of 77% in persistent endodontic infections [6]. The ability of *E. faecalis* to form endodontic biofilm has been well studied and established [7,8,9]. Biofilm formation provides *E. faecalis* with better protection to environmental threats as well as enhanced tolerance to antimicrobials [10]. Literature reports the presence of *E. faecalis* biofilm in medicated root canals and its survival in conditions of severe alkaline stress (pH = 11.2) [7,11]. In addition, the complexity of the root canal system (isthmus, lateral canal and apical ramification) complicates traditional chemo-mechanical debridement and makes biofilm removal challenging [3,12,13].

Typically, different irrigant solutions are used between instrumentation during orthograde as well as retrograde endodontic therapy [14], with the goal of removing bacteria in the endodontic lumen, dissolving the smear layer, and disinfecting the canals. Sodium hypochlorite (NaOCl) used alone or in combination with other disinfectants, such as EDTA, citric acid, chlorhexidine etc., represents the most used endodontic irrigant. Notwithstanding the effective disinfectant potential of NaOCl, it is not able to completely remove the smear layer created during instrumentation and cases of reinfection have been observed over time.

In an attempt to enhance the effect of irrigants, provide a biofilm-free surface and make the endodontic treatment more predictable, sonic techniques (<20,000 Hz) have been introduced [15,16,17]. It was noteworthy that the sonic technique alone was unsuccessful in biofilm eradication from the root canal surface [16]. Further attempts to remove biofilm completely have been made and include investigation of ultrasonic (>20,000 Hz) effect in combination with various irrigants [18,19,20]. This decontamination strategy has been shown to remove the intracanal biofilm more efficiently when compared to conventional methods of irrigation [21]. Still, similar to sonic techniques, no complete biofilm removal could be achieved by using the ultrasonic system [22].

In the early 2000s, the application of Er:Yag lasers was suggested as a means of disinfecting root canal space [23]. Since then, studies focused on the investigation of the efficacy of the Er:Yag laser family in biofilm removal. It is considered that the bactericidal potential of Er:Yag laser is related to the evaporation effect of cellular water, which expands quickly during the laser pulse and leads to the disintegration of bacterial cell wall [16,24,25,26].

Another emerging technique, photon-induced photoacoustic streaming (PIPS), which is provided by Er:Yag lasers, has shown promising results in providing the removal of the biofilm from the root canal surface [15,27]. This technique implies positioning of the PIPS tip inside the pulp chamber with a consequent activation of irrigants inside the root canal through a profound photoacoustic and photomechanical phenomenon. Each impulse created by the PIPS tip is absorbed by the water molecules, further creating a strong “shock wave” that leads to the formation of an effective streaming of fluids inside the canal while avoiding side effects, such as high temperature [28].

The contemporary biofilm removal techniques, such as sonic-, ultrasonic- and Er:Yag laser-assisted techniques, have become increasingly popular over the last years. Consequently, this paper aimed to provide an overview of the published work discussing in vitro endodontic biofilm cultivation methods associated with sonic-, ultrasonic- and laser-assisted removal techniques. Furthermore, the effectiveness of these removal techniques was discussed and an overview of the evaluation of the cleaning methods was provided.

## 2. Materials and Methods

An extensive literature search of articles was performed by two investigators (U.J. and C.M.) using the electronic databases PubMed and Scopus. The following keywords and strings were used: ultrasonic OR YSGG OR YAG OR ER OR Er,Cr AND biofilm AND root canal OR endodontics. No time restrictions were set, and the only filter applied was articles published in the English language. The last search was conducted in February 2022 and it yielded 122 titles. Abstracts were read and excluded if the reported article did not have any possible applications in endodontic biofilm removal achieved by the sonic, ultrasonic and Er:Yag devices. Finally, 41 articles were fully read and included in the present paper.

## 3. Results and Discussion

Findings from the included articles were sorted into the following sections:Biofilm cultivation;Biofilm removal techniques;Evaluation of biofilm removal.

### 3.1. Biofilm Cultivation

#### 3.1.1. Monospecies Biofilm

The endodontic bacteria are usually organized in biofilm communities which are present not only in the main canal, but in the overall root canal system [29]. The extracellular matrix of the biofilm offers bacteria higher survival rates in challenging growth and environmental conditions [11]. In order to mimic conditions that are well established within infected root canals, authors put their efforts into growing *E. faecalis* biofilms since it is considered to be the leading pathogen associated with failed endodontic treatment. Among the microorganisms commonly isolated in the endodontic space, this microorganism represents the leading pathogen largely associated with failed endodontic treatment [7].

According to the reviewed articles, potential double origins, lab-adapted strains [15,16,18,19,20,21,24,26,30,31,32,33,34,35,36,37,38,39,40,41,42,43,44,45,46] and clinically isolated *E. faecalis* were noted [25]. Types of *E. faecalis* strains which are most widely used by different authors and bacteria origin are shown in Table 1. *E. faecalis* strain isolated from root canal of pulpless teeth is available, but only two authors reported using it [24,30], while the majority of the studies used strains of E. faecalis isolated from different tissues and fluids.

When choosing the strain of *E. faecalis*, dental researchers should be aware of its genetic heterogeneity which is observed inside a single population, as well as that different strains of *E. faecalis* can be detected in the oral cavity of one individual [47,48].

#### 3.1.2. Multispecies Biofilm

Mixed endodontic infections are more common than infections caused by a single microorganism [11]. Collecting subgingival plaque from dental patients is a method used to grow multispecies biofilm [17,49]. Another approach for multispecies biofilm sampling is an intraoral contamination process by wearing a custom-made appliance for 6–8 days. However, this method is very subjective as the patients voluntarily carry the appliance and follow the diet recommendations [49]. A simplified, lab-adapted, dual-species biofilm model of *E. faecalis* and *Streptococcus mutans* was introduced, as well as a three-species biofilm composed of *E. faecalis*, *Streptococcus mitis* and *Campylobacter rectus* [43,50]. Only one study investigated the removal of dual-species biofilm composed of vancomycin resistant *E. faecalis* and *Candida albicans* [35]. Niazi et al. (2014) used a biofilm consisting of five different species of microorganisms (Table 1) [51].

#### 3.1.3. Biofilm Mimicking

Few studies included in this paper used non-bacterial approaches to test different methods of biofilm removal. Macedo et al. (2014) proposed a hydrogel model to provide visualization of biofilm removal by ultrasonic techniques. As stated by the author, viscoelastic properties of hydrogel can be compared to the one of bacterial biofilm and therefore it may be suitable for replacing bacterial biofilm in in vitro studies [52]. Joy et al. (2015) applied layers of stained collagen to the dentin surface and analyzed digital images of its removal by ultrasonic irrigation [22].

#### 3.1.4. Substrate and Period of Incubation

The majority of research included in this paper uses human [15,16,18,19,21,22,24,25,27,30,31,32,33,34,35,37,38,40,41,42,43,44,45,46,49,53,54] and animal-bovine dentin [20,55] as substrate for biofilm growth and formation (Table 1).

A general pattern in preparing samples for bacterial inoculation was observed among the studies which included both human and bovine dentin: after examining the extracted teeth, root canals were enlarged and shaped using endo files, with sodium hypochlorite (NaOCl) serving as an irrigant and EDTA used for smear layer removal. In contrast, Meire’s et al. (2012) presented a different approach since the crowns were firstly cut and dentin slices of standardized thickness were obtained for further testing [38]. Similarly, Bao et al. (2017) used a split tooth model which, after biofilm removal efforts have been made, allows dissembling the tooth and gaining a clear insight into the dentin surface [53]. Another methodology observed in the reviewed studies focuses on use of bovine dentin sections that serve as a substrate for multispecies biofilm cultivation. These sections were incorporated within an orthodontic device and worn by a volunteer allowing oral bacteria to accumulate on the dentin surfaces [55].

Hydroxyapatite (HA) discs are frequently used in dental research and possess the affinity towards bacteria colonization [56,57]. Consequently, both Noiri et al. (2008) and Shen et al. (2010) used hydroxyapatite discs for *E. faecalis* biofilm cultivation [17,26].

Further, six studies included in this review used root canal models as substrate for biofilm formation. In the most recent studies, root canal models were created using CAD technology and 3D printing. The goal of 3D printing is to create a desirable, transparent and anatomically standardized model which would allow an insight into real-time interaction between irrigants and biofilm removal [39,58].

Time plays an important role in biofilm formation, allowing bacteria to aggregate and form a network of polymer strands. Scanning electron microscope (SEM) investigations revealed that after 1 week of incubation, a biofilm-like structure can be observed on dentin surface. After 2, 3 and 4 weeks, biofilm becomes thicker and thus more challenging to remove. Mature biofilm with characteristic honey-comb like structures can be observed after 6 weeks of incubation [9].

The period of incubation used for biofilm cultivation is presented in Table 1. As seen from the table, no consensus in terms of incubation period between different authors was found. However, Cheng et al. (2012, 2017) and Mohhmed et al. (2016, 2017) were consistent in choosing the same incubation period for their studies over the course of years [24,25,39,58]. The average period of incubation for the reviewed studies was 14 days for multi- and 17.8 days for monospecies biofilm, with major variation between 1 and 50 days.

In studies reviewed in this paper, the authors determined the incubation period based on data available from the literature and their personal preference. However, as pointed out earlier, various incubation periods result in different maturity and thickness of the biofilm, which eventually can lead to unequal effort towards biofilm removal.

### 3.2. Biofilm Removal Techniques

#### 3.2.1. Sonic Devices

Table 2 shows the sonic devices, settings and irrigant solutions found during the literature search. Among the different sonic devices, the EndoActivator was the most used apparatus. According to its manufacturer, deep cleaning of the root canal system and subsequent biofilm removal could be expected.

Sodium hypochlorite (NaOCl) is one of the most commonly used irrigants in endodontic practice. Authors are in agreement that sonic energizing with different concentrations of NaOCl offers greater biofilm disruption than sonic energizing with water or saline [15,33]. Furthermore, sonic energizing with NaOCl was found to be an effective and promising technique in biofilm reduction in many different studies. [15,16,39,58]. Maden et at. (2017) developed a prototype device which using low electric current is able to sonically agitate the NaOCl solution. This device was able to significantly reduce biofilm in comparison to other sonic devices [37].

Chlorhexidine (CHX) is also a popular irrigant due to its antimicrobial effect [59]. It has been shown that the antimicrobial effect of sonic irrigation with 2% chlorhexidine was superior when compared to sonic saline irrigation. Additionally, it was concluded that longer exposure time to irrigants (up to 3 min) and use of CHX–Plus contributed to higher number of dead bacterial cells [17].

Alternative irrigants used in the reviewed studies were microbubble-emulsion (ME) and QMiX solution. Halford et al. (2012) examined the synergistic effect of ME and sonic agitation. This combination provided bacteria reduction 3 mm from the apical terminus, but left a considerable number of viable bacteria 1 mm from the apical terminus [60]. Interestingly, EndoActivator in combination with QMiX solution provides more favorable antibiofilm efficacy than NaOCl needle irrigation. However, as stated by the authors of the study, this result may also be due to chemical properties of QMIX solution in which the detergent plays an important role in weakening the biofilm structure [15].

#### 3.2.2. Ultrasonic Devices

Passive ultrasonic irrigation (PUI) is a term used in endodontics for describing irrigation of root canal system without additional shaping of the canal wall [61]. With the intention to avoid possible confusion and misunderstanding, PUI will be referred to as “ultrasonic irrigation” in further text. In contrast to previously discussed studies where EndoActivator is the most commonly used sonic device, authors used different units in an attempt to enhance biofilm removal by ultrasonic agitation of irrigants. Table 3 shows reported details of ultrasonic agitation when investigating biofilm removal efficacy. Non-consistent power settings of ultrasonic devices, various shapes, and sizes of ultrasonic tips, different irrigant concentrations and time of irrigation used in research, make comparison of the studies and their findings quite problematic and prone to subjective interpretation. Nevertheless, we aimed to summarize findings from the reviewed studies, based on similarities observed in their methodology.

With the aim of investigating purely mechanical effects of ultrasonic devices, only saline or distilled water was used during biofilm removal (Table 3). Ultrasonic agitation of saline had proven to be more efficient in multispecies biofilm removal than simple irrigation with saline delivered by syringe and needle. This result can be due to pure mechanical effect of the ultrasonic agitation, since no antibacterial agent was used [50]. The results are in agreement with a similar research [32], that reported bacterial reduction using a comparable approach in monospecies biofilm elimination. Similarly, Grundling et al. (2011) and Hartmann et al. (2019) stated that ultrasonic irrigation with distilled water offers significant biofilm reduction when compared to manual agitation of saline with hand files. Furthermore, this study was based on a microscopy evaluations (SEM) method and confirmed a significant difference in apical and middle thirds between manually agitated saline and ultrasonic irrigation with distilled water [20,62].

NaOCl can also be used as an irrigant during ultrasonic agitation. Bhuva et al. (2010) demonstrated that ultrasonic irrigation with NaOCl is superior to saline needle/syringe irrigation in biofilm removal at all three levels of the root canal [19]. Comparatively, other studies noted similar results, although the evaluation method of biofilm removal was different and included plate counting (CFU method) [41,44,45]. In addition, it was shown that ultrasonic NaOCl irrigation offers better bacterial reduction than ultrasonic irrigation with water or saline, which can be explained by the antimicrobial effect of NaOCl [30,33]. Both the ultrasonic device and GentleWave system were effective in reducing the bacteria inside the root canal space [63].

Similarly to NaOCl, CHX can be ultrasonically agitated. Cherian et al. (2016) investigated the effectiveness of ultrasonic agitation of CHX and compared it to CHX syringe irrigation. It was concluded that ultrasonically delivered CHX provides significant bacterial reduction in comparison to syringe CHX irrigation [21]. Furthermore, Shen at al. (2010) compared the antimicrobial efficacy of CHX with CHX-Plus, both ultrasonically agitated, and found a significant difference in the number of cells killed. CHX-Plus was more efficient in biofilm reduction, which can be contributed to the chemistry of the antimicrobial agent itself [17]. Yet, when observing the study, it should be noted that HA discs were used as substrate for multispecies biofilm formation, which is notably different when compared to the morphology of the root canal system. Similarly, when activated ultrasonically, enzymes are more efficient in biofilm removal compared to saline alone [51].

Lastly, ultrasonic effect within simulated biofilm and root canal models was also investigated. For this purpose, Macedo et al. (2014) introduced a transparent root canal model with isthmus and lateral canals which were filled with hydrogel. As a result, the main canals were better cleaned with water used as an irrigant rather than NaOCl. Different from lateral canals, isthmi were equally well rinsed regardless of the agent used for ultrasonic irrigation [52]. Another study used root canal models to investigate fluid dynamics generated by syringe irrigation and both continuous and intermittent ultrasonic technique [36]. Continuous ultrasonic agitation was found to be significantly better in biofilm removal compared to syringe irrigation and intermittent ultrasonic technique. The superior action of continuous ultrasonic agitation can be due to complete oscillating amplitude of the ultrasonic tip inside the root canal which, consequently, generates maximum acoustic microstreaming. Unlike complete oscillating amplitude achieved by the continuous tip, the intermittent ultrasonic tip comes in occasional contact with the canal wall, thus resulting in weakened microstreaming effect [36]. Very recently, Mohmmed et al. explored the effect of different agitation methods using NaOCl as irrigant within 3D printed root canals [39]. The results indicated an effective biofilm removal with NaOCl ultrasonic agitation especially when compared to sonic and syringe irrigation. Additionally, microscopic images evaluations showed that 1 mm from the apex manual and sonic treatment left the biofilm intact, while complete biofilm removal at the same level was associated with ultrasonic agitation of NaOCl [39,58].

#### 3.2.3. Er:Yag Laser Group

##### Er:Yag Laser

In one of the pioneer studies which investigated the effect of Er:Yag laser in biofilm removal, Noiri et al. (2008) directly irradiated hydroxyapatite discs that had previously been contaminated with multispecies biofilm. As shown in Table 4, different energy pulses were applied and it was discovered that low laser energy offers anti-biofilm effect [26]. Although the study demonstrated encouraging results in biofilm removal, a notably different scenario in clinical conditions may be found, since the laser tip is not able to reach all parts of the complex root canal anatomy. Meire et al. (2012) used a rather similar laboratory approach, although the authors evaluated the effect of Er:Yag laser in monospecies biofilm removal and used dentin discs as substrate. Additionally, NaOCl was introduced as an irrigant, and it was concluded that the combination of Er:Yag irradiation and NaOCl irrigation can be used as joint techniques during root canal disinfection since it provides better biofilm removal than Er:Yag laser alone [38].

However, similarly to the previous study, a uniform irradiation of dentin discs was possible due to the laboratory setup of the experiment. Complex root canal morphology in clinical conditions represents a greater challenge in biofilm removal, but, nonetheless, the results of the mentioned studies confirm beneficial effect of laser irradiation in attempts to remove biofilms.

A more relevant clinical approach was proposed by Cheng et al. (2012) who compared the results of biofilm removal using different techniques and irrigants. Although conventional 5.25% NaOCl syringe irrigation of canals was effective in eliminating the bacteria from the surface of root canals, CFU counting revealed it was not able to successfully remove *E. faecalis* from deep dentin layers. By applying Er:Yag laser and NaOCl as irrigant, better biofilm reduction was achieved deep inside dentinal tubules, thus suggesting that Er:Yag laser supports penetration of NaOCl. Furthermore, the study emphasized the importance of the synergistic effect of NaOCl Er:Yag laser agitation since it showed better results in comparison to NaOCl syringe irrigation or saline Er:Yag agitation [24].

##### Er,Cr:YSGG Laser

The pure effect of Er,Cr: YSGG laser on biofilm removal without the presence of irrigant/dry canal was investigated by Cheng et al. (2012) [24]. Using laser parameters as shown in Table 5, it was concluded that Er,Cr: YSGG laser is less effective than 5.25% NaOCl irrigation. Furthermore, the same study revealed that Er:Yag NaOCl agitation is superior to Er,Cr: YSGG laser irradiation alone.

Surprisingly, other studies found no difference in biofilm removal between Er,Cr: YSGG 4% NaOCl agitation and 4% NaOCl syringe irrigation [33]. On the other hand, Seet et al. (2012) discovered that Er,Cr: YSGG 4% NaOCl agitation offers better biofilm eradication compared to 4% NaOCl syringe irrigation [16]. Interestingly, both authors used the identical E. faecalis strain, same period of incubation and the same irrigant concentration and time of agitation. Even though Seet et al. (2012), Betancourt et al. (2019) and Suer et al. (2020) used lower laser power settings compared to Chriso et al. (2016), they still found superior results in biofilm removal which were associated with Er,Cr: YSGG laser agitation of the irrigant, rather than conventional syringe irrigation [46,64].

##### PIPS

The goal of PIPS is to enhance biofilm removal by creating photoacoustic shockwaves that would travel through the root canal system which is filled with an irrigant [65]. When applying the PIPS technique, the laser tip is usually positioned in the access cavity (pulp chamber or canal entrance). Many authors are consistent in their methodologies with an emphasis that, during studies, the position of the tip was limited to the access cavity only, without further insertion towards the root canal [27,30,54,55]. Instead, De Meyer et al. (2017) inserted the PIPS tip into the canal, 6 mm short of the working length, only to discover equal effect of PIPS, regardless of the position of the laser tip [50].

As seen from Table 5, the same laser parameters, i.e., pulse rates from 10 to 20 Hz and pulse energies from 10 to 40 mJ, which were considered to have no thermal or ablative effect on canal walls, were used during investigation of PIPS effect in biofilm reduction [15,27,30,35,54,55]. However, different laser tip designs as well as various irrigant concentrations and activation times are noted in the reviewed articles (Table 5).

In general, the PIPS technique is considered to be superior to conventional syringe/needle irrigation, regardless of the irrigant used [15,25,30,43,50,54]. Confocal laser scanning microscopy images taken by Al Shahrani et al. (2014) revealed that conventional NaOCl irrigation leaves viable bacteria deep inside dentinal tubules, while PIPS with NaOCl offers deeper penetration of the irrigant, consequently killing more bacteria [30].

When comparing PIPS to sonic agitation, Ordinola-Zapata et al. (2014) demonstrated that PIPS significantly reduces the number of bacteria within bovine root canal models when NaOCl was used as irrigant [54]. Contrarily, Balic et al. (2016) and Hage et al. (2019) concluded that both PIPS and sonic irrigation of NaOCL remove biofilm evenly from the root canal [15,66].

Up to the present time, it has been confirmed that, compared to ultrasonic techniques, PIPS offers enhanced biofilm removal in the apical part of root canals [49]. SEM images from different studies confirm PIPS superiority over ultrasonic methods in biofilm reduction inside root canals [55]. Moreover, by evaluating treatment results by CLSM and CFU, Nelaakantan et al. (2015) concluded that PIPS agitation of NaOCl and etidronic acid provides better biofilm removal when compared to conventional and ultrasonic techniques with the same irrigants [40]. Furthermore, PIPS was more efficient than sonic devices in removing hydrogel from the isthmus when using only water as irrigant [67].

Only one study compared the effect of PIPS to Er,Cr:YSGG laser in dual-species biofilm removal. The study used saline as irrigant and therefore it was possible to estimate solely the physical effect of lasers. Er,Cr:YSGG laser agitation of non-antimicrobial agent performed better at E. faecalis and C. albicans biofilm removal in comparison to PIPS [35].

Lastly, Golob et al. (2017) suggested a modified PIPS protocol, which offered promising results in disinfection of root canals [27]. Unlike the classic PIPS protocol, the authors introduced PIPS with EDTA, prior to NaOCl irrigation, and removed the mineralized part of the smear layer, opening dentinal tubules, thus enabling deeper penetration of NaOCl. Additionally, in order to increase the safety of the PIPS treatment, laser energy was reduced by 50% and no difference was found in biofilm removal between higher and lower power settings.

### 3.3. Evaluation of Biofilm Removal

The most frequently used methods for evaluating biofilm removal efficacy include counting of colony forming units (CFU) and analysis of scanning electron microscope (SEM) images, while confocal laser scanning microscopy (CLSM), polymerase chain reaction (PCR) and transmission electron microscopy (TEM) are found to be less mentioned in the reviewed studies. Additionally, it was found that some authors used more than one means of evaluation while assessing the success of biofilm removal [18,21,24,26,30,40,45] (Table 1).

#### 3.3.1. CFU—Plate Counting

Methods of obtaining samples for further microbial analysis differ among the studies. It is suggested that after treatment protocol, root canals are filled with sterile saline, followed by syringe aspiration, centrifugation and counting of CFU [42]. Similarly, paper points leave the integrity of the dentin surface intact and have also been used in collecting samples for bacteriological evaluation [24,30]. On the other hand, Hedstrom files [32], round dental [34], Gates Gliden burs [21,40] and Peeso reamer [41] allowed researchers to retrieve dentin samples from various depths and use them for later analysis. Regardless of the sampling technique used, the CFU method provides information on the number of viable bacteria found either on the root canal surface or at various dentin depths.

#### 3.3.2. SEM

An innovative proposal introduced by Bhuva et al. (2010) involves SEM image observation and analysis by endodontists with different levels of experience [19]. Briefly, a scoring system was created in relation to percentage of root canal which was covered with biofilm and dentists rated the SEM images according to their personal opinion and observations. A similar approach in SEM analysis was also used a few years later by Bhardway et al. (2013) and Ordinola-Zapata et al. (2014) [31,55]. Eventually, dividing the root canal surface into three areas—coronal, middle and apical—and taking SEM images of the mentioned sections is also widespread in methodologies reviewed by this paper [16,19,20]. The observed level of magnification used varies, ranging from 40 up to 10.000× [19,21,30,39,53]

Overall, SEM allows visualization of morphological structures of biofilms, their amount and distribution on dentin surface, as well as in deeper dentin layers. [20,24] However, it should be noted that sample preparation for SEM analysis might result in changes of the biofilm’s extracellular polymer matrix [11].

#### 3.3.3. CLSM

Based on the reviewed papers, it was noted that the main advantage of using CLSM techniques is the author’s capability to distinguish viable and dead cells within biofilms. When observing the CLSM images taken after the treatment, live cells are usually seen as green, while dead cells are painted red [17,30,39]. Additionally, 3D reconstruction can be achieved and the ratio between live and dead cells can also be determined [40].

#### 3.3.4. Other Methods

TEM: Mohmmed et al. (2017) used TEM as well as CLSM and SEM to evaluate the results of biofilm removal. TEM images enabled an insight into cellular integrity and level of damage caused by different removal techniques [39].

PCR: Only two authors used PCR with the purpose of confirming identification of E. faecalis and to determine the presence of bacteria even in low numbers or stationary phase and therefore avoiding false negative results [15,18]. Quantitative real time PCR analysis was also reported as a valid method for the evaluation of the bacterial removal [63]. However, one should keep in mind that even DNA from dead cells, as well as free extracellular DNA, could be amplified and detected, eventually giving misleading data.

Histology: Brown–Brenn staining technique can be used to determine bacterial penetration into dentinal tubules. Only one study was found to use this technique, with an observational magnification set to 100× and 400× [49].

High speed camera: One study used a high speed camera attached to a microscope to record the hydrogel removal process from transparent root canal models. The recorded films of hydrogel removal were analyzed in MATLAB and later discussed [52]. A similar methodology was used by Mohmmed et al. (2016), although this study investigated the removal of E. faecalis biofilm in contrast to the previously described study that investigated removal of biofilm mimicking model [58].

Colorimetric assay: Layton et al. (2015) used colorimetric assay, a rapid technique for biofilm quantification, while assessing the results of root canal cleaning. Additionally, by using micro PIV system, this research provided valuable findings concerning irrigation and fluid dynamics in simulated root canals [36].

Digital images: A special imaging method that allows estimation of the canal surface which is covered with collagen was introduced by Joy et al. (2015). As stated by the author, the method used in this study allows the three-dimensional irregularities on the root canal surface to become two-dimensional surfaces on the images [22].

It is important to emphasize that the evaluators should always be blinded to the treatment protocol in all techniques that employ image analysis.

## 4. Future Research

Cultivating a biofilm formation in in vitro conditions may seem easily replicable, low cost and offering researchers control over the period of incubation and maturity of biofilms introduced in studies. However, one should consider that different growth media, different conditions and time of incubation lead to various viscoelastic behaviors of biofilms, which is ultimately an important feature for the resistance of biofilms.

In the reviewed articles, the models used for establishing biofilm infection were straight, single-rooted and single-canal or simulated canal. In order to provide more relevant clinical implications and draw comparison of the results from the in vitro trials more accessible, the following strategies should be considered:standardize the pathogens’ growth conditions which can lead to more uniform viscoelastic properties of biofilms and their thicknesses;confirm biofilm formation by SEM/CLSM before initiating the treatment protocol;introduce root canals with complex morphology to surveys;align sonic, ultrasonic and laser parameters, respectively, and standardize them;besides CFU, introduce SEM and CLSM, which would allow a more detailed insight into the effectiveness of disinfection methods in the coronal, middle and apical parts of root canal, as well as the distinction between live and dead bacterial cells.

## 5. Conclusions

Within the limitations of this study, it can be concluded that sonic, ultrasonic and Er:Yag laser agitation, in general, offer better biofilm removal when compared to conventional irrigation methods delivered by syringe and needle. The choice of the right irrigation solution is an important factor for removal of the endodontic biofilm, with water and saline being less effective compared to NaOCl and CHX. However, due to heterogeneity in methodologies, it is difficult to compare adjuvant endodontic techniques with one another and give recommendations for the most efficient method in biofilm removal. Lastly, this review emphasizes the importance of standardizing methodologies in experimental protocols, as well as introducing strategies which would provide more relevant clinical implications.

## Figures and Tables

**Table 1 polymers-14-01334-t001:** Details of the biofilm cultivation methods and cleaning evaluation techniques of the reviewed studies. CFU: colony forming unit; SEM: scanning electron microscope; CLSM: confocal laser scanning microscope; PCR: polymerase chain reaction; TEM: transmission electron microscope.

Author, Year	Microorganism	Period of Incubation (Days)	Substrate	Methodology Assessment
Noiri et al. (2008)	*E. faecalis* ATCC 19246	21	HA disc	CFU, SEM
Shen et al. (2010)	Subgingival plaque	21	HA disc	CLSM
Bhuva et al. (2010)	*E. faecalis* OMGS 3202	3	Human dentin	SEM
Alves et al. (2011)	*E. faecalis* ATCC 29212	30	Human dentin	CFU, PCR
Peters et al. (2011)	Oral bacteria	6–8 intraorally, 15 *in vitro*	Human dentin	CFU, histology
Grundling et al. (2011)	*E. faecalis* ATCC 29212	50	Animal teeth	SEM
Meire et al. (2012)	*E. faecalis* ATCC 10541	1	Human dentin	CFU
Case et al. (2012)	*E. faecalis* ATCC 29212	12	Human dentin	CFU
Halford et al. (2012)	*E. faecalis* ATCC 29212	7	Human dentin	CFU
Cheng et al. (2012)	*E. faecalis* ATCC 4083	28	Human dentin	CFU and SEM
Seet et al. (2012)	*E. faecalis* ATCC 700802	28	Human dentin	SEM
Bhardway et al. (2014)	*E. faecalis* ATCC 29212	3	Human dentin	SEM
Niazi et al. (2014)	*E. faecalis* OMGS 3202, *Propionibacterium acnes, Staphylococcus epidermidis, Actinomyces radicidentis, Streptococcus mitis*	14	Hydroxyapatite discs	CFU, CLSM
Ordinola-Zapata et al. (2014)	Oral biofilm	3 days intraorally, 2 days *in vitro*	Animal dentin	SEM
Macedo et al. (2014)	Biofilm mimicking with hydrogel	/	Solidifying polydimethylsiloxane	High-speed camera
Al Shahrani et al. (2014)	*E. faecalis* ATCC 4083	21	Human dentin	CFU, SEM
Olivi et al. (2014)	*E. faecalis* vancomycin-resistant	28	Human dentin	SEM
Nelaakantan et al. (2015)	*E. faecalis* ATCC 29212	21	Human dentin	CFU, CLSM
Layton et al. (2015)	*E. faecalis* ATCC 29212	21	PEG-modified PDMS	crystal violet assay
Chirsto et al. (2016)	*E. faecalis* ATCC 700802	28	Human dentin	CFU
Joy et al. (2016)	Biofilm mimicking with collagen	/	Human dentin	Digital images
Balic et al. (2016)	*E. faecalis* ATCC 29212	15	Human dentin	PCR, CFU
Pladisai et al. (2016)	*E. faecalis* ATCC 29212	21	Human dentin	CFU
Mohmmed et al. (2016)	*E. faecalis* ATCC 19433	10	Clear liquid photopolymer material	fluorescence microscope with high-resolution CCD camera
Cherian et al. (2016)	*E. faecalis* ATCC 29212	7	Human dentin	CFU, SEM
De Meyer et al. (2017)	*E. faecalis* (strain ATCC 10541) *Streptococcus mutans* (strain LMG 14558)	2	Resin	CFU
Toljan et al. (2017)	*E. faecalis* ATCC 29212	1	Human dentin	CFU
Bao et al. (2017)	Mixed biofilm	28	Human dentin	SEM
Mohmmed et al. (2017)	*E. faecalis* ATCC 19433	10	Clear liquid photopolymer material	SEM, CLSM, TEM
Kasic et al. (2017)	*E. faecalis* *Candida albicans*	7	Human dentin	CFU
Cheng et al. (2017)	*E. faecalis* (clinically isolated)	28	Human dentin	SEM
Golob et al. (2017)	*E. faecalis* vancomycin-resistant	28	Human dentin	SEM
Maden et al. (2017)	*E. faecalis* ATCC 29212	21	Human dentin	CFU
Betancourt et al. (2018)	*E. faecalis* ATCC 29212	1	Glass	CFU and atomic force microscope
Sasanakul et a. (2019)	*E. faecalis* ATCC 29212	21	Human dentin	CFU
Zhang et al. (2019)	Sub- and supragingival biofilm	14	Human dentin	Quantitative real-time PCR
Hartmann et al. (2019)	*E. faecalis* ATCC 19433	26	Human dentin	CFU
Suer et al. (2020)	*E. faecalis* ATCC 29212	1	Human dentin	SEM
Hoedke et al. (2021)	*E. faecalis* ATCC 29212 and *Streptococcus oralis* ATCC 35037	5	Human dentin	CFU
Choi et al. (2021)	*E. faecalis* OG1RF ATCC 47077, *Streptococcus mitis* ATCC 49456 and *Campylobacter rectus* ATCC 33238	21	Human dentin	CFU, CLSM, TEM
Afkhami et al. (2021)	*E. faecalis* ATCC 29212	28	Human dentin	CFU

**Table 2 polymers-14-01334-t002:** Details on irrigants, mode and time of agitation, as well as type of sonic device used for biofilm removal.

Author	Sonic Device	Irrigant	Mode of Agitation	Time of Agitation
Shen et al. (2010)	Endo Activator Advanced Endodontics, Santa Barbara, CA, USA	2% chlorhexidine digluconate (CHX), CHX plus	Medium power	1–3 min
Halford et al. (2012)	Endo Activator Dentsply Tulsa Dental Specialties, Tulsa, OK, USA	Sterile water, 5.25% NaOCl, or microbubble emulsion	Full energy	20 s
Seet et al. (2012)	Endo Activator Dentsply (Maillefer, Ballaigues, Switzerland)	Saline, 4% NaOCl	Full energy	60 s
Balic et al. (2016)	Endo Activator Dentsply, Maillefer, Ballaigues, Switzerland	2.5% NaOCl and QMiX solution	10,000 cpm	30 s
Mohmmed et al. (2016)	Endo Activator Dentsply Tulsa Dental Specialities, Tulsa, OK, USA	2.5% NaOCl	High power	30 s
Mohmmed et al. (2017)	Endo Activator Dentsply Tulsa Dental Specialities, Tulsa, OK, USA	2.5% NaOCl	High power	30 s
Maden et al. (2017)	Endo Activator Dentsply Tulsa Dental Specialities, Tulsa, OK, USA	5.25% NaOCl	167 Hz	60 s
Swimberghe et al. (2019)	Eddy (VDW) and EA (Dentsply Sirona, Konstanz, Germany)	Water	6000 Hz	60 s
Hoedke et al. (2021)	SONICflex, (KaVo, Warthausen, Germany)	Saline, 1% NaOCl	Intensity mode 3	60 s

**Table 3 polymers-14-01334-t003:** Details on type of ultrasonic devices and instruments (manufacturers), irrigant, mode and time of agitation used in the studies included in the review.

Author	Ultrasonic Device	Irrigant	Mode	Time of Agitation	Instrument
Shen et al. (2010)	E7 of Varios 350 LUX (Nakanishi Inc., Kanuma, Japan)	Saline, 2% CHX, CHX-plus	Medium power	60–180 s	Ultrasonic tip
Bhuva et al. (2010)	Piezon Master 400 (Electro Medical Systems SA, Nyon, Switzerland)	1% NaOCl	¼ of maximum power	40 s	Size #15 ultrasonic file
Alves et al. (2011)	Piezoelectric (Enac- Osada, Tokyo, Japan)	2.5% NaOCl, 0.2% CHX	Not specified	60 s	Size #15 K-file
Peters et al. (2011)	EMS 600 ultrasonic (Nyon, Switzerland)	6% NaOCl	5/10 of maximum power	30 s	Non-cutting insert
Grundling et al. (2011)	Nac Plus ultrasonics (Adiel, Ribeirao Preto, SP, Brazil)	Distilled water, 2% NaOCl	Scale power 2	Not specified	Size #40 K-file
Case et al. (2012)	Ultrasonic scaler (Perioscan; Sirona, Bensheim, Germany)	Saline	70 kHz and 200 mW/cm^2^	120 s	Size #15 K-file
Halford et al. (2012)	P5 Newtron unit (Acteon Group, Norwich, UK)	Sterile water, 5.25% NaOCl, microbubble emulsion	Power setting 10	60 s	Size #10 K-file
Bhardway et al. (2014)	Ultrasonic unit (Satelec, Merignac Cedex, France)	1% NaOCl	¼ of maximum power	40 s	Size #15 ultrasonic file
Ordinila-Zapata et al. (2014)	Satelec P5 suprasson ultrasonic unit (Suprasson P5; Satelec Acteon group, Acteon, Merignac, France)	6% NaOCl	Power setting 4	60 s	Irrisafe file 20.00 (Acteon, Merignac, France)
Niazi et al. (2014)	Ultrasonic unit (Piezon Master 400; Electro Medical Systems, Nyon, Switzerland)	Trypsin, Proteinase K, NaOCl, saline, CHX	¼ of the maximum power	20 s	15 ultrasonic file (Endosonore File, Dentsply Maillefer)
Macedo et al. (2014)	Ultrasonic device (Suprasson P-Max, Acteon Satelec, Acteon, Merignac, France)	Water and 8.7% NaOCl	Power setting ‘Yellow 5’	20 s	IrriSafe file (Acteon, Merignac, France)
Layton et al. (2015)	1. ultrasonic device (P5 Newtron unit; Satelec); 2. PiezoFlow device (Dentsply Tulsa Dental Specialties, Konstanz, Germany)	Sterile water	1. power setting 10; 2. power setting 5	20 s	1. non-cutting steel wire, 200 μm; 2. ultrasonic irrigation needle, 500 μm
Nelaakantan et al. (2015)	EMS 600 ultrasonic unit (Nyon, Switzerland)	Saline, 6 and 3% NaOCl, 18% etidronic acid, 17% EDTA	Not specified	30 s	Ultrasonic file
Joy et al. (2015)	Not specified	2.5% NaOCl	Not specified	Not specified	Size #15 K-file
Pladisi et al. (2016)	Piezoelectric ultrasonic device (P5; Satelec Acteon, Merignac, France)	2.5% NaOCl	Power setting 4	60 s	Irrisafe tip K20/21 (Acteon, Merignac, France)
Toljan et al. (2016)	Ultrasonic device (Piezon Master 400; EMS, Nyon, Switzerland)	3% NaOCl	Medium power	30 s	Size #15 K-file
Cherian et al. (2016)	Ultrasonic unit (Varios 750, NSK Nakanishi Inc., Tochigi, Japan.)	2% CHX, 0.1% octenidine dihydrochloride	¼ of maximum power	40 s	Ultrasonic file size 15
Mohmmed et al. (2016)	Satelec P5 Newtron piezon unit (Acteon, Merignac, France)	2.5% NaOCl	Power setting 7	30 s	Irrisafe instrument 20/02 (Acteon, Merignac, France)
De Meyer et al. (2017)	Ultrasonic device (Suprasson Pmax Newtron, Satelec)	Sterile saline, 2.5% NaOCl	50% power mode	20 s	Irrisafe file size 20 (Acteon, Merignac, France)
Bao et al. (2017)	Ultrasonic device (ProUltra, Dentsply Tulsa Dental, Konstanz, Germany)	3% NaOCl	Power setting 3	60 s	U-file size 20
Mohmmed et al. (2017)	Satelec P5 Newtron piezon unit (Acteon, Merignac, France)	2.5% NaOCl	Power setting 7	30 s	Irrisafe file size 20/02 (Acteon, Merignac, France)
Betancourt et al. (2018)	Newtron P5 XS, Satelec (Acteon, Merignac, France)	Saline, 0.5% and 5% NaOCl	Medium power	60 s	Stainless steel 25/00, 25 mm in length
Hartmann et al. (2019)	D700 Dabi Atlante (Ribeirao Preto, SP, Brazil)	Saline, 17% EDTA, 0.5% peracetic acid	Not specified	60 s	Ultrasonic files size 15 (Mani Inc., Utsunomiya Tochigi, Japan)
Zhang et al. (2019)	ProUltra PiezoFlow Active Ultrasonic System	3% NaOCl, 8% EDTA, sterile water	Not specified	320 s	Not specified
Swimberghe et al. (2019)	P5 Newton; Satelec (Acteon, Merignac, France)	Water	Power 7	60 s	Size 25 (Irrisafe; Satelec Acteon, Merignac, France)
Hoedke et al. (2021)	Not specified	Saline, 1% NaOCl	30% power	60 s	Size 25 IRRI S file VDW
Choi et al. (2021)	Satelec P5 Newtron XS ultrasonic unit (Acteon, Merignac, France)	Water, 1% NaOCl	Power 6	30 s	Irrisafe (Acteon, Merignac, France); CK (B&L Bio, Ansan, Korea); Endosonic Blue (Maruchi, Wonju, Korea)

**Table 4 polymers-14-01334-t004:** Details on laser type, irrigant, laser tip design and parameters, time of irradiation and position of the tip used for biofilm removal in the studies included in the review.

Author	Laser Type	Irrigant	Laser Tip	Laser Parameters (Wavelength, Power, Pulse Energy, Pulse Frequency, Pulse Duration)	Time	Position of the Tip
Noiri et al. (2008)	Er:YAG laser (Arwin; MORITA, Osaka, Japan)	No irrigant	Custom made tip, diameter 650 μm	2940 nm not specified 20, 40, 80 mJ not specified	10 s	3 mm from the HA disc
Meire et al. (2012)	Er:Yag laser (Fidelis; Fotona, Ljubljana, Slovenia)	0.25% NaOCl	RO2 handpiece (Fotona)	2940 nm not specified 50, 100 mJ 15 Hz not specified	20 s	Directly over the dentin disc
Cheng et al. (2012)	1. Er:Yag laser (Fontona Lasers) 2. Er,Cr:YSGG laser (Biolase, Irvine, CA)	1. 5.25% NaOCl, saline, distilled water 2. Not specified	1. Optical fiber, 200 μm diameter 2. Optical fiber, 415 μm diameter	1. 2.940 nm 0.3 W not specified 15 Hz not specified 2. 2.780 nm 1 W not specified 20 Hz not specified	1. 20 s 2. 60 s	1. Orifice of the root canal 2. 1 mm from the working length
Seet et al. (2012)	Er,Cr:YSGG laser (WaterLase, Biolase Technology, Irvine, CA, USA)	Saline, 4% NaOCl	Radial firing tip (17 mm, 52°)	Not specified 0.25 W not specified 20 Hz not specified	60 s	4 mm into the canal, withdraw coronally
Christo et al. (2016)	Er,Cr:YSGG laser (Waterlase, Biolase Technology, Irvine, CA, USA)	Saline, 0.5, 1 and 4% NaOCl	RFT 3 (diameter 415 μm, length 17 mm) (Endolase, Biolase Technology)	2.780 nm 0.5 W 25 mJ 20 Hz 140 μs	60 s	5 mm apically from the orifice
Kasic et al. (2017)	Er,Cr:YSGG laser (Waterlase, Biolase, Irvine, CA, USA)	Saline	RTF 2 (200 μm)	Not specified 1.25 W not specified 15 Hz 150 μs	Not specified	5 mm apically from the coronal access
Betancourt et al. (2018)	Er,Cr:YSGG laser (Waterlase iPlus BIOLASE technology, Irvine, CA, USA)	Saline, 0.5% and 5% NaOCl	RFT 2 tip (Endolase, BIOLASE Technology, Inc.; 200 μm in diameter, length 21 mm, calibration factor of >0.55)	2.780 nm, 1 W, 100 mJ 10 Hz 140 μs	60 s	Tip placed in the cylindric reservoir
Suer et al. (2020)	Er,Cr:YSGG laser	2.5% NaOCl	Fiber tip	Not specified 2 W/0.75 W Not specified 20 Hz	40 s	Placed into the canal towards the apex

**Table 5 polymers-14-01334-t005:** Details on laser type, irrigant, laser tip design and parameters, time of irradiation and position of the tip used during PIPS.

Author	Laser Type	Irrigant	Laser Tip	Laser Parameters (Pulse Rate, Pulse Energy, Pulse Duration, Power)	Time	Position of the Laser Tip
Peters et al. (2011)	Er:Yag laser Fidelis; (Fotona, Ljubljana, Slovenia)	3.6% NaOCl	21-mm-long, 400-μm endodontic fiber	10 Hz 50 mJ not specified not specified	30 s	Coronal reservoir
Olivi et al. (2014)	Er:Yag laser (LightWalker AT, Fotona, Ljubljana, Slovenia)	5% NaOCl followed by 17% EDTA	9-mm, 600-μm quartz tip	15 Hz 20 mJ 50 μs 0.3 W	90 s	Access cavity
Ordinola-Zapata et al. (2014)	Er:Yag laser Fidelis; (Fotona, Ljubljana, Slovenia)	6% NaOCl	12-mm, 400-μm quartz tip	15 Hz 20 mJ 50 μs 0.3 W	60 s	Access cavity
Al Shahrani et al. (2014)	Er:Yag laser (LightWalker AT, Fotona, Ljubljana, Slovenia)	6% NaOCl, saline	9-mm, 600-μm quartz tip	15 Hz 20 mJ 50 μs 0.3 W	90 s	Access cavity
Neelakantan et al. (2015)	Er:Yag laser (Fidelis; Fotona, Ljubljana, Slovenia)	Saline, NaOCl-EDTA-NaOCl, NaOCl-EDTA, NaOCl-editronic acid	21 mm long, 400 microns endodontic conical fiber tip	10 Hz 50 mJ 50 ms not specified	30 s	Coronal reservoir
Balic et al. (2016)	Er:Yag laser (LightWalker AT, Fotona, Ljubljana, Slovenia)	2.5% NaOCl, QMiX solution	600-μm fiber tip	15 Hz 20 mJ 50 μs not specified	60 s	Access cavity
Kasic et al. (2017)	Er:Yag laser (LightWalker AT, Fotona, Ljubljana, Slovenia)	Saline	14-mm, 400 μm tapered tip	15 Hz 20 mJ 50 μs 0.3 W	40 s	Access cavity
Cheng et al. (2017)	Er:Yag laser (Fotona, Ljubljana, Slovenia)	0.5 and 5.25% NaOCl	PIPS tip (diameter 300 μm, Fotona)	25 Hz 20 mJ 50 μs 0.5 W	30 s	1 mm below the orifice of the canals
Golob et al. (2017)	1. Er:Yag laser Fidelis; (Fotona, Ljubljana, Slovenia) 2. Er:Yag laser (LightWalker AT, Fotona, Ljubljana, Slovenia)	1, 3, 5% NaOCl,EDTA, sterile water	PIPS tip (600/9)	1. 15 Hz 20 mJ 50 μs not specified 2. 15 Hz 10 mJ 50 μs not specified	30 s	Access cavity
De Meyer et al. (2017)	Er:Yag laser (AT Fidelis, Fotona, Ljubljana, Slovenia)	2.5% NaOCl, saline	14 mm, 400 μ, fiber tip	20 Hz 20 and 40 mJ, 50 μs not specified	20 s	Canal entrance, root canal
Swimberghe et al. (2019)	Er:Yag-laser (Lightwalker; Fotona, Ljubljana, Slovenia)	Water	PIPS 400/14	20 Hz 20 mJ 20 μs Not specified	60 s	Canal entrance
Hage et al. (2019)	Er:Yag-laser (Lightwalker; Fotona, Ljubljana, Slovenia)	Water	PIPS tip (9 mm long; 600 μm diameter)	15 Hz 20 mJ 50 μs not specified	Not specified	Canal entrance
Afkhami et al. (2021)	Er:Yag-laser (Lightwalker; Fotona, Ljubljana, Slovenia)	Suspension of AgNP, 5% NaOCl	PIPS tip	15 Hz, 20 mJ 50 μs 0.3 W	30 s	Pulp chamber

## Data Availability

Not applicable.

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
