# Peer review of "Biofilm in Endodontics: In Vitro Cultivation Possibilities, Sonic-, Ultrasonic- and Laser-Assisted Removal Techniques and Evaluation of the Cleaning Efficacy"

_polymers, 2022, doi:10.3390/polym14071334_

Round 1
Reviewer 1 Report
The manuscript "Biofilm in Endodontics: in Vitro Cultivation, Contemporary Eradication Techniques and Evaluation of The Cleaning Efficacy" by Uros Josic et al. is well written and contains a lot of detailed information organised in Tables. In my opinion the work is valuable, but it contains a few minor errors that should be corrected:
- Line 29: "intraradicular" - Shouldn't be interradicular?
- Line 31: "Gram positive" - Please change to "Gram-positive".
- Please check the spelling of "E. faecalis" and other bacterial/fungal names in the text. All species names should be written in italics.
- Line 76: "... was performed by..." - In my opinion, this detail is irrelevant here.
- Please check the numbering of the particular sections. For example, "Biofilm cultivation" (line 90) should be numbered as a subsection of "3. Results and Discussion" (line 85), so instead of marking it as "1", please start with "3.1".
- Line 50: Unnecessary dot can be spotted before references [15-17].
- In the paragraph (lines 93-109) the bacterial name E. faecalis is mentioned too often. Please try to change some of them to other expressions to avoid repetitions.
- In the Table 1 C. albicans name is mentioned for the first time. However, in the whole text there is no full name of the fungus: Candida albicans. Please correct it. The same should be done for E. faecalis (Enterococcus faecalis) and other strains when they are mentioned for the first time in the text.
Reviewer 2 Report
Dear Authors, the review entitled Biofilm in: in Vitro Cultivation, Contemporary Eradication Techniques and Evaluation of The Cleaning Efficacy summarize actual knowledge about biofilm eradication in endodontics. The review is correctly composed and mostly focused on physicochemical methods for endodontics biofilm eradication.
Comments;
- Authors wrote that “Over the past decade, biofilm eradication techniques, such as sonication of irrigant solutions, ultrasonic and laser devices have been investigated in laboratory settings” However also other approaches were investigated such as the use of specific enzymes eg.
Niazi SA, Clark D, Do T, Gilbert SC, Foschi F, Mannocci F, Beighton D. The effectiveness of enzymic irrigation in removing a nutrient-stressed endodontic multispecies biofilm. Int Endod J. 2014
- The tables formatting should be done with omitting vertical lines.
- The graphical abstract or block schema that will summarize the review could significantly improve its quality.
Reviewer 3 Report
The goal of this review is to present the in vitro models for assessing biofilm removal efficacy of different endodontic techniques. The topic is relevant and up-to-date. The manuscript presents interesting information, however, the organization of the information needs improvement.
First, the title is too generic for the content of the manuscript. There are various contemporary eradication techniques used in endodontics that are not addressed in this review. Therefore, the title needs to be adjusted to the techniques addressed in the present manuscript.
In line with the previous comment, the last paragraph of the introduction, with the objectives of the study, needs to be re-written, to make clear for the readers what they can expect from the present review. Do authors want to explore all the techniques available to deal/remove biofilm from root canals or only some specific type of techniques?. Also the rationale followed by the authors to choose some current methods and ignore others eventually with similar efficacy must be presented at introduction section.
Material and methods should present inclusion and exclusion criteria for the review, search protocol and define which type of review is intended with this manuscript.
Some additional comments:
Last column of table 1, consider the title “Methodology of assessment”;
P6L126- There are at least 2 additional studies using a dual-species of E. faecalis and C. albicans to evaluate the efficacy of different endodontic protocols to remove biofilm:
https://doi.org/10.3389/fmicb.2017.00498 and
https://doi.org/10.1016/j.pdpdt.2018.04.009
The section of evaluation of biofilm removal, specially subsection 3.4. Other methods, needs some critical appraisal about the utility/advantage od using such methodologies of assessment.
Round 2
Reviewer 3 Report
Authors addressed superficially some comments of the first review. The title and abstract need to be adjusted with the real objectives of the present manuscript.
In the present version, authors state at the abstract: “This review aimed to give a comprehensive overview of endodontic biofilm cultivation methods, eradication techniques and, lastly, methods used for the evaluation of cleaning efficacy”. As I comment in the first review, the stated objective is clearly wider than the scope of the present review, therefore, goals need to be adjusted in the abstract and also the present title still needs improvement.
This review is limited to sonic, ultrasonic and LASER assisted methods, which are a limited group of strategies to remove endodontic biofilm. In this sense, for the sake of clarity and rigor the goals stated by the authors need to be redefined.
Moreover, the in vitro biofilm study models presented are limited to papers published with the sonic, ultrasonic and LASER assisted methods, excluding all the other biofilm models used to test other methods for endodontic biofilm removal. Therefore, the present title is misleading, as it suggests that this paper will provide an overview of all models of biofilm in endodontics. For this reason, adjustment of the title is advisable, because additional biofilm endodontic models have been used and published to assess other methodologies developed to manage endodontic biofilm.
The number of studies obtained from Pubmed with this search protocol need to be presented and also the number of papers excluded “if the reported article did not have any possible applications in endodontic biofilm removal achieved by the sonic, ultrasonic and Er:Yag devices.”
Future research section: what is the meaning of “3. introduce curved canals to surveys”?
Author Response
Please, see the attachment.
Thank you.

Round 3
Reviewer 3 Report
Corrections performed.